# Binary Combinations of Essential Oils: Antibacterial Activity Against *Staphylococcus aureus*, and Antioxidant and Anti-Inflammatory Properties

**DOI:** 10.3390/molecules30030438

**Published:** 2025-01-21

**Authors:** Clara Naccari, Giovanna Ginestra, Nicola Micale, Ernesto Palma, Benedetta Galletta, Rosaria Costa, Rossella Vadalà, Antonia Nostro, Mariateresa Cristani

**Affiliations:** 1Dipartimento di Scienze della Salute, Università “Magna Græcia” di Catanzaro, Viale Europa, 88100 Catanzaro, Italy; c.naccari@unicz.it (C.N.); palma@unicz.it (E.P.); 2Dipartimento di Scienze Chimiche, Biologiche, Farmaceutiche ed Ambientali, Università degli Studi di Messina, Viale F. Stagno D’Alcontres 31, 98166 Messina, Italy; benegalletta@yahoo.it (B.G.); anostro@unime.it (A.N.); mcristani@unime.it (M.C.); 3CIS—Centro Servizio Interdipartimentale—IRC-FSH “Centro di Ricerche Farmacologiche, Sicurezza degli Alimenti e Salute ad Alto Contenuto Tecnologico”, Università “Magna Græcia” di Catanzaro, Viale Europa, 88100 Catanzaro, Italy; 4Fondazione “Prof. Antonio Imbesi”, Università degli Studi di Messina, Piazza Pugliatti 1, 98122 Messina, Italy; 5Dipartimento di Scienze Biomediche, Odontoiatriche e delle Immagini Morfologiche e Funzionali, Università degli Studi di Messina, Via Consolare Valeria, 98100 Messina, Italy; costar@unime.it (R.C.); rvadala@unime.it (R.V.)

**Keywords:** essential oils, antibacterial activity, antioxidant activity, anti-inflammatory activity, binary combinations, FIC index, animal infections

## Abstract

*Background*: The lack of new antimicrobial drugs and increased antimicrobial resistance has focused attention on the employment of essential oils (EOs) in human and veterinary medicine. The aim of this study was to test new binary associations between known and uncommon EOs. *Methods*: EOs from *Origanum vulgare* L., *Juniperus communis* L., *Cistus ladaniferus* L., *Citrus aurantium* L. var. *amara* were tested individually and in binary combinations to study, as follows: antibacterial activity against *Staphylococcus aureus*, including methicillin-resistant *S. aureus* (MRSA) and *Escherichia coli*; antioxidant capacity via redox-based assays (DPPH, ABTS and FRAP); and anti-inflammatory activity via the bovine serum albumin denaturation inhibition assay. *Results: O. vulgare* L. showed good antibacterial activity against all strains (MIC = 0.03–0.12%, *v*/*v*), followed by *C. ladaniferus* L., and also had the best antioxidant and anti-inflammatory activities. Synergistic and additive effects were observed for the EO combinations *O. vulgare* L./*C. ladaniferus* L. and *O. vulgare* L./*J. communis* L. against *S. aureus* and MRSA, respectively. A reduction in biofilm was noted. Antioxidant and anti-inflammatory activities were also detected. *Conclusions*: The results suggest that EO combinations may be a promising strategy in veterinary settings for the treatment of infectious diseases caused by *S. aureus*, including drug-resistant and biofilm-forming strains accompanied by oxidative stress and inflammation.

## 1. Introduction

The current lack of new antibacterial drugs, coupled with the continued rise in antimicrobial resistance (AMR), poses a critical problem for global health. To effectively address this challenge requires a One Health approach, an integrated strategy that recognizes the connection of human, animal, and environmental health [1,2].

Combating AMR is critical for several reasons, as follows: to safeguard human health, to prevent the emergence and spread of antibiotic-resistant bacteria; to preserve the effectiveness of antimicrobials used in human and veterinary medicine; and to minimize the presence of antibiotic residues in animal-derived food products.

Given the increasing emergence of drug-resistant bacteria responsible for infections in animal farms, such as mastitis, mammary pustular dermatitis and skin infections, there is an urgent need to explore natural and alternative therapeutic approaches.

In recent years, the attention of many researchers has focused on the study of natural products as a support to conventional antibiotic therapy. Among various plant-derived secondary metabolites, essential oils (EOs) are complex mixtures of volatile compounds, primarily terpenes, along with aldehydes, alcohols, and esters [3]. Numerous studies have documented the antimicrobial, antifungal, antioxidant, and anti-inflammatory properties of various EOs, with a particular emphasis on species originating from the Mediterranean region [4]. Among these, EOs from *Cistus ladaniferus* L., *Citrus aurantium* L. var. *amara*, *Juniperus communis* L.,and *Origanum vulgare* L., are of traditional and new interest due to their known and well-documented biological properties, including antibacterial, antioxidant, anti-inflammatory, analgesic, antispasmodic, angiogenic, antiplatelet, antimutagenic and antigenotoxic properties [4,5].

*Cistus ladaniferus* L. (Cistaceae family) shows potential hypoglycemic, hypolipidemic, antihypertensive activities [6]. *Citrus aurantium* L. var. *amara* (Rutaceae family), a major crop in Mediterranean regions, is also used as a sedative due to its soothing and calming effects [7], and as a natural anticonvulsant agent [8]. *Juniperus communis* L. (Cupressaceae family), widely distributed across Europe, Asia, and North America, is used as a diuretic and for digestive disorders [9], and also exhibits hypoglycemic, hypolipidemic, and hepatoprotective properties [10]. *Origanum vulgare* L. (Lamiaceae family), one of the most used medicinal plants in infections of the respiratory tract, is also a valid antiparasitic, digestive and antispasmodic agent; in addition, it has demonstrated hypoglycemic effects [11].

The study of the pharmacological activities of EOs is also of growing interest in veterinary medicine. There are various in vitro studies in the literature on their efficacy against common bacteria, such as *Escherichia coli* and *Staphylococcus aureus*, which are often responsible for mastitis in bovine animals and sheep, for mammary pustular dermatitis in ovine animals [12], and for exudative epidermitis in pigs, etc. [12,13,14,15]. EOs have also shown efficacy against bacteria isolated from the milk samples of mastitic sheep [16] and pathogens isolated from animal-derived products [17].

However, it is necessary to underline that, due to their chemical complexity, although most EOs are generally considered safe, their use could result in the development of both local or systemic toxic effects such as skin and allergic reactions, and nephrotoxic, hepatotoxic and cancerogenic effects [18]. Therefore, combinations of natural EOs may be a valid strategy to reduce minimum active doses, moderate possible adverse side effects, and also enhance their efficacy against resistant pathogens responsible for infectious diseases. 

However, although the potential therapeutic properties of EOs are well known, there are limited data on their biological activities when used in combinations.

The aim of this study was to evaluate the in vitro antibacterial, antioxidant and anti-inflammatory properties of the EOs of *Cistus ladaniferus* L. (Cistus), *Citrus aurantium* L. var. *amara* (Bitter orange), *Juniperus communis* L. (Juniper), and *Origanum vulgare* L. (Oregano) used in binary combinations. Their potential synergistic or additive effects for possible use against *S. aureus*, a clinically significant Gram-positive pathogen responsible for common animal infections, were explored.

## 2. Results

### 2.1. GC-MS Chromatographic Analysis

The detailed composition of *Citrus aurantium* L. var. amara L., *Cistus ladaniferus* L., *Juniperus communis* L., and *Origanum vulgare* L. EOs were obtained through GC-MS and are reported in Appendix A.

### 2.2. Antibacterial Activity

The MIC and MBC values of the EOs tested in our study are reported in Table 1. Among the EOs, *O. vulgare* L. revealed the best inhibitory activity against *S. aureus* and *E. coli*, with MIC values of 0.03–0.12% *v*/*v* and MBC values of 0.06–0.25% *v*/*v*, respectively, followed by *C. ladaniferus* L., which exhibited a moderate antibacterial activity, with MIC values between 0.25% and 0.5% *v*/*v*.

#### 2.2.1. Checkerboard Assay

To study whether *O. vulgare* L. in combination with the other EOs resulted in a higher degree of bacterial inhibition, a checkerboard assay was performed. The results of the antibacterial activity of the EO combinations are given in Table 2.

The fractional inhibitory concentration (FIC) was calculated according to the following formulas: FICA = MICA + B/MICA, FICB = MICB + A/MICB; and FIC Index = FICA + FICB, where MICA + B is the MIC of compound A in presence of compound B, and MICB + A is the opposite. The FIC Index (FICI) values were interpreted as follows: synergistic effect for FICI ≤ 0.5; additive effect for FICI > 0.5–1; indifference for FICI > 1 to <2; and antagonism for FICI ≥ 2 [19].

The three tested combinations (*O. vulgare* L./*C. ladaniferus* L., *O. vulgare* L./*C. aurantium* L. var. *amara*, and *O. vulgare* L./*J. communis* L.) showed synergistic or additive effects (FICI: 0.312–1) against all tested bacterial strains except *E. coli*, for which they showed an indifferent effect (FICI: 1.25–1.5). Specifically, a synergy (FICI from 0.312 to 0.50) with a 4- to 16-fold reduction in MIC values was recorded for *S. aureus* ATCC 6538 and an additive effect (FICI from 0.625 to 1) was shown for the MRSA strain. No antagonistic effect was observed.

In terms of the concentrations, the combinations that displayed a synergistic effect against *S. aureus* ATCC 6538 were, as follows: *O. vulgare* L./*C. ladaniferus* (L. 1/4 + 1/4 of sub-MICs corresponding to 0.0078/0.0625% *v*/*v*); *O. vulgare* L./*C. aurantium* L. var. *amara* (1/4 + 1/8 of sub-MICs corresponding to 0.0078/0.125% *v*/*v*); and *O. vulgare* L./*J. communis* (L. 1/4 + 1/16 of sub-MICs corresponding to 0.0078/0.0625% *v*/*v*). The combinations with additive effects against the MRSA strain were, as follows: *O. vulgare* L./*C. ladaniferus* L. (1/2 + 1/8 of sub-MICs corresponding to 0.0625/0.0625% *v*/*v*); *O. vulgare* L./*C. aurantium* L. var. *amara* (1/4 + 1/2 of sub-MICs corresponding to 0.0312/0.5% *v*/*v*); and *O. vulgare* L./*J. communis* L. (1/2 + 1/2 of sub-MICs corresponding to 0.0625/0.5% *v*/*v* of *J. communis* L.).

Figure 1 shows representative images of the isobolograms of *O. vulgare* L. in combination with the other EOs against *S. aureus* ATCC 6538 and MRSA ATCC 43300.

#### 2.2.2. Effect on Biofilm Formation

The OD_492_ of *S. aureus* ATCC 6538 biofilm control stained with safranin was 1.50 ± 0.15. Regarding the anti-biofilm effect, the sub-synergistic concentrations of 1/4 + 1/4 EO combinations resulted in a good reduction in the *S. aureus* ATCC 6538 biofilm compared to that of the control (Figure 2).

A good inhibition of biofilm formation with the 1/2 synergistic concentration was observed for all combinations. In particular, biofilm reductions of 80% for *O. vulgare* L./*C. ladaniferus* L. (corresponding to 0.0039/0.0312% *v*/*v*), 75% for *O. vulgare* L./C. *aurantium* L. var. *amara* (corresponding to 0.0039/0.125% *v*/*v*), and 68% for *O. vulgare* L./*J. communis* L. (corresponding to 0.0039/0.125% *v*/*v*) were detected. Interestingly, a good inhibitory effect (75% reduction in biofilm formation) was maintained in the presence of the 1/4 synergistic combination of *O. vulgare* L./*C. ladaniferus* L. (corresponding to 0.0019/0.0156%, *v*/*v*).

### 2.3. Antioxidant Activity

In order to characterize the antioxidant properties of the EOs studied, DPPH, ABTS and FRAP assays were chosen among the different validated benchmark methods. These redox-based assays measure the reducing capacity of the tested samples under specific conditions. The results of the antioxidant activity assays of the individual EOs are summarized in Table 3.

All EOs showed a medium antioxidant/free radical scavenger activity. The potency order in all assays was higher for *O. vulgare* L., followed by *J. communis* L., *C. ladaniferus* L., and *C. aurantium* L. var. *amara*. For all the assays, the antioxidant activity of the EOs was compared to that of the positive control (Trolox for the DPPH and ABTS assays and Fe_2_SO_4_ for the FRAP assay).

### 2.4. Anti-Inflammatory Activity

The ability to inhibit the denaturation of bovine serum albumin (BSA) was calculated because proteic denaturation is recognized as a source of inflammation and its inhibition is a key indicator of anti-inflammatory potential [20,21].

Significantly higher activity was also observed in the BSA assay from the EO of *O. vulgare* L. compared with the other EOs studied. The results are expressed as IC_50_ and are presented in Figure 3.

### 2.5. Antioxidant and Anti-Inflammatory Activities of EO Combinations

The EO combinations (*O. vulgare* L./*C. ladaniferus* L., *O. vulgare* L./*C. aurantium* var. *amara* L. and *O. vulgare* L./*J. communis* L. (1:1 *v*/*v*)) were evaluated for their antioxidant activity by DPPH, ABTS, and FRAP assays and for their anti-inflammatory activity by a BSA denaturation assay. The results are presented in Table 4 and Figure 3, respectively.

The combinations analyzed for antioxidant activity (Table 5) demonstrated an additive effect in the ABTS and FRAP assays with *O. vulgare* L./*J. communis* L. An additive effect was also observed in the DPPH and FRAP assays with *O. vulgare* L./*C. ladaniferus* L. The other combinations were indifferent. None of the combinations displayed antagonistic effects. This clearly demonstrates that variability exists between the studied methods, and that the employment of different assays, as in this study, provides a better overall assessment of efficacy.

Similar kinds of interactions were demonstrated in the BSA assay for anti-inflammatory activity (Table 6); the combinations of *O. vulgare* L./*J. communis* L. and *O. vulgare* L./*C. ladaniferus* L. showed additive effects.

## 3. Discussion

The EOs from *C. ladaniferus* L., *C. aurantium* L. var. *amara, J. communis* L. and *O. vulgare* L., alone and in binary combinations, demonstrated antibacterial, antioxidant and anti-inflammatory properties, suggesting that these could be natural agents for potential therapeutic use in the treatment of animal infectious diseases, including those caused by drug-resistant and biofilm-forming bacteria. Furthermore, they may attenuate the oxidative stress and inflammation that often accompany such infections.

The biological activities of these EOs are related to their chemical composition [22], and, in particular, to major components including, as follows: carvacrol for *O. vulgare* L.; α-pinene for *J. communis* L.; camphene and α-pinene for *C. ladaniferus* L.; and limonene for *C. aurantium* L. var. *amara* [23,24,25].

Regarding the antibacterial properties of individual EOs, *O. vulgare* L. showed a good inhibitory activity against *S. aureus* and *E. coli*, followed by *C. ladaniferus* L. In contrast, *J. communis* L. and *C. aurantium* L. var. *amara* exhibited lower efficacy against these bacterial strains. The results obtained for *O. vulgare* L. are in line with those of other authors, who highlighted the antibacterial activity of EO against both Gram-positive and Gram-negative strains [26,27]. Antimicrobial activity has also been reported for *C. ladaniferus* L. and *J. communis* L. EOs [6,28]. Concerning *C. aurantium* L. var. *amara*, the antimicrobial potential and the impact of seasonal variations on the chemical composition and biological activities of its EO have been exhaustively described [29].

The results obtained from the DPPH, ABTS and FRAP tests showed good antioxidant/free radical scavenging capacity for all EOs examined individually. The antioxidant effects of EOs are derived from their ability to neutralize free radicals by donating hydrogen atoms or electrons, thus protecting biological molecules from oxidative damage. Although the (poly)phenolic constituents are mainly responsible for these properties, other compounds, such as cyclic monoterpenes and various functional groups, also make an important contribution. Therefore, the combined presence of these diverse components (as in this case; see hereinafter) improves the overall antioxidant activity of EOs [22,30].

With regard to anti-inflammatory activity, the BSA denaturation assay revealed favorable responses from the tested EOs, with *O. vulgare* L. having the highest efficacy among all the samples. This assay is based on the principle that the denaturation of proteins leads to the loss of their structural integrity and function, resulting in the potential production of auto-antigens. The bioactive compounds present in EOs may protect against this process by preserving the various bonds involved in maintaining the protein structure. This protective effect is the basis of the observed anti-inflammatory activity of the EOs studied [20]. The superior biological activities detected in the EO of *O. vulgare* L. are generally attributable to its major component, i.e., carvacrol [31,32,33]. However, the effectiveness of *O. vulgare* L. can be influenced by its other minor components such as *p*-cymene, γ-terpinene, thymol and (E)-caryophyllene [4].

*O. vulgare* L. was chosen as the main EO for the study of binary combinations because it showed the best response in the assays that were carried out. Overall, our investigations on the antibacterial, antioxidant, and anti-inflammatory properties of binary combinations (*O. vulgare* L./*J. communis* L., *O. vulgare* L./*C. ladaniferus* L. and *O.vulgare* L./*C. aurantium* L. var. *amara*) revealed noteworthy interactions, which were quantified using FIC values.

The effects of all tested combinations were synergistic against *S. aureus*, additive against MRSA, and indifferent against *E. coli*. In terms of concentrations, the optimal combination was found to be *O. vulgare* L./*C. ladaniferus* L. against *S. aureus* ATCC 6538 (0.0078% *v*/*v* of *O. vulgare* L. and 0.0625% *v*/*v* of *C. ladanifer* L.) and MRSA (0.0625% *v*/*v* of *O. vulgare* L. and 0.0625% *v*/*v* of *C. ladaniferus* L.). Interestingly, the EO combinations did not reduce the effectiveness of single EOs, as no antagonistic effects were observed. *S. aureus* is a bacterium that can cause a wide range of infections, from minor skin conditions to severe systemic diseases. Its resistance to multiple antibiotics has led to the emergence of MRSA, a major public health concern. Additionally, *S. aureus* is known for its ability to form biofilms, which enhances its persistence and resistance to treatments. Consequently, addressing *S. aureus* infections, especially those caused by drug-resistant strains, remains a critical area of research. Our data contribute valuable insights into the effectiveness of the EOs analyzed, providing further information on their potential application in combating MRSA.

The subsequent study conducted on *S. aureus* biofilm also demonstrated the best inhibitory effect (80% reduction in biofilm formation) through the synergistic association between *O. vulgare* L./*C. ladaniferus* L. These results highlight the potential of combining EOs against bacterial biofilms, which are notoriously difficult to treat. Biofilms are complex communities of microorganisms that adhere to surfaces and are embedded in a self-produced extracellular matrix. They are very difficult to eradicate due to their poor susceptibility to conventional antimicrobial agents and host immune defense [34].Therefore, the development of new strategies able to inhibit *S. aureus* biofilm formation is of great interest, considering the ability of this bacterium to cause several diseases.

With respect to antioxidant activity, an additive effect was observed for all the binary combinations of EOs under evaluation, whereas in the BSA denaturation assay, an additive effect in the anti-inflammatory response was evident only for the combinations *O. vulgare* L./*J. communis* L. and *O. vulgare* L./*C. ladaniferus* L. Our findings pertaining to antioxidant activity are in line with those of other research conducted on different combinations of EOs [35,36]. For example, some studies have noted the synergistic effects of combinations of EOs from the Laminaceae family (except *O. vulgare* L.) including, as follows: *Apium graveolens* L.; *Thymus vulgaris* L. and *Coriandrum sativum* L. [37]; *Thymus fontanesii Boiss. & Reut.*, *Artemisia herba-alba* Asso and *Rosmarinus officinalis* L. [38]; and *Callistemon lanceolatus* Sweet, *Ocimum gratissimum* L., *Cymbopogon winterianus Jowitt* ex Bor., *Cymbopogon flexuosus* (Nees ex Steud.) Stapf, *Mentha longifolia* (L.) L. and *Vitex negundo* L. [39].

Regarding the anti-inflammatory properties of EO combinations, there are few studies in the literature [40]. Some researchers have reported on the effect of associations between EOs and common anti-inflammatory drugs [41,42]. However, there is growing interest in the study of the synergistic anti-inflammatory effects of combined phytochemicals [43].

The results of the interactions between the binary associations of EOs in all the assays performed (antibacterial, antioxidant and anti-inflammatory) require further consideration. The observed synergistic effect, defined as the combined effect of the tested compounds greater than the sum of the individual effects [44], could be mainly due to the composition of the EOs, which can affect multiple biochemical processes, enhance the bioavailability of the components, and/or neutralize the adverse effects [39]. On the other hand, an additive effect is considered as the resulting effect of two EOs equal to the sum of the individual effects. For example, for the EO of *O. vulgaris* some authors have observed a synergistic effect in association with EO of *Rosmarinus officinalis* L. and additive effects in association with EOs of *Thymus vulgaris* L., *Ocimum basilicum* L. and *Origanum majorana* L. [45].

In general, the synergistic and additive antibacterial effects of the binary associations could be attributed to the main component of *O. vulgare* L. EO, namely, carvacrol, and its remarkable effects on the structural and functional properties of the cytoplasmatic membrane [46,47]. Carvacrol could increase the permeability of the cytoplasmic membrane and facilitate the entry of the other EO components into the cell. However, α-pinene can contribute to the structural damage of the cell membrane [48] and limonene can alter cytoplasmic membrane permeability [49]. As for camphene, several studies have documented the antibacterial activity of this terpene and its derivatives [50], particularly as a potential inhibitory agent against *S. aureus* [51].

Similarly, in terms of antioxidant activity, carvacrol may contribute to the additive effect resulting from the binary combinations of EOs assessed in the different assays.

With respect to anti-inflammatory activity, the additive effects observed for the *O. vulgare* L./*J. communis* L. and *O. vulgare* L./*C. ladaniferus* L. combinations could be due to the main component of *O. vulgare* L. EO [52]. In fact, carvacrol was able to reduce the production of inflammatory mediators (IL-1β, IL-4, IL-8, malondialdehyde, and prostanoids) [23] and the induction of IL-10 release [53]; α-pinene reduced the production of inflammatory cytokines (IL-1β, NF-κB, and LTB4) [54]. In contrast, limonene was shown to increase IL-10 levels and reduce TNF-α levels [55]. With respect to camphene, it is possible to hypothesize its contribution towards lipoxygenase inhibition, as documented by other authors for the EO of *Cistus albidus* [56].

The results of this study suggest that combinations of EOs rich in bioactive components may enhance their overall pharmacological effects, particularly in terms of their antibacterial, antioxidant, and anti-inflammatory properties.

It is known that EOs, despite being natural compounds, can cause toxic effects. It is difficult to define a real safety profile for EOs; therefore, the possibility of combining them in binary associations may allow for a reduction in the dose required for activity and thus reduce the risk of potential side effects.

In the current study, this approach offers a promising strategy for improving the management of animal infectious diseases caused by *S. aureus,* including infections from drug-resistant and biofilm-forming strains that are often accompanied by oxidative stress and inflammation.

Further studies on these EO combinations are worthy of further evaluation to better understand the advantages of their potential applications.

## 4. Materials and Methods

### 4.1. Essential Oils Sampling

Four commercially available EOs were purchased from two different Italian companies. *C. ladaniferus* L. (Cistus) was purchased from Laborbio-Collegno (Torino, Italy) and *C. aurantium* L. var. *amara* (Bitter orange), *J. communis* L. (Juniper), and *O. vulgare* L. (Oregano) were purchased from FLORA srl-Lorenzana (Pisa, Italy). 

### 4.2. GC-MS Chromatoghraphic Analysis

The GC-MS analyses were carried out on a GCMS-TQ8030 system (Shimadzu, Milan, Italy) equipped with an AOC-20i auto-sampler. Samples of *Citrus aurantium* L. var. amara EO were injected neat; the injection volume was 0.4 μL with a split ratio of 1:50 at 250 °C. *Cistus ladaniferus* L., *Juniperus communis* L., and *Origanum vulgare* L. EOs were preliminarily diluted to 1:5 *v*/*v* in chloroform; the injection volume was 1.0 μL with a split ratio of 1:50 at 250 °C. The capillary column was an SLB-5ms (Supelco, Milan, Italy) with a 30 m × 0.25 mm ID × 0.25 μm film thickness, operated at the following oven program: 50 °C (2 min), increased to 250 °C (held for 10 min) ≅4 °C/min. The mass spectrometric source (EI) was set at 200 °C, 0.95 kV; the interface was 250 °C and the acquisition mode was in full scan with a range of 40–350 *m*/*z* and a scan speed of 1666 amu/sec. Data handling was performed by means of GCMSsolution software 2.4 (Shimadzu). For peak assignment, the following mass spectral libraries were used: FFNSC 2, Adams 4th edition, Wiley 9, NIST11, NIST webbook. n-Paraffins (C7–C40, custom-made mixture) were injected apart from real samples in order to measure the retention indices. Peak identification was based on the library-matching of unknowns (similarity index ≥ 90) and retention index matching of the experimental vs. the published values (RI filter ± 10 units) [57].

### 4.3. Antibacterial Activity

#### 4.3.1. Bacterial Strains and Culture Conditions

The following strains were used: *Staphylococcus aureus* ATCC 6538; methicillin-resistant *Staphylococcus aureus* (MRSA) ATCC 43300; and *Escherichia coli* ATCC 10536. Cultures for the antimicrobial tests were grown at 37 °C in Mueller–Hinton Broth (Millipore Sigma, Darmstadt, Germany) for 24 h.

#### 4.3.2. MIC and MBC Determination

The minimum inhibitory concentration (MIC) and the minimum bactericidal concentration (MBC) of the EOs were determined using a broth dilution micro-method in 96-well round-bottomed polystyrene microtiter plates according to the guidelines of the Clinical and Laboratory Standards Institute [58], with some modifications for EOs. Briefly, *C. ladaniferus* L., *C. aurantium* L. var. *amara*, *J. communis* L. and *O. vulgare* L. EOs were dissolved to 50% using dimethylsulfoxide (DMSO). Then, serial twofold dilutions were made in MHB at concentrations ranging from 1% to 0.0039% *v*/*v*. The DMSO maximum concentration was 0.5% (*v*/*v*). Bacterial cultures were inoculated to yield a final concentration of 5 × 10^5^ CFU/mL and 5 × 10^4^ CFU/well, approximately. A growth control (medium with DMSO and without EO) and a positive control (ciprofloxacin) were included. Plates were incubated at 37 °C for 24 h. The MIC was considered as the lowest concentration of the EO giving the inhibition of visible bacterial growth after incubation for 24 h. To evaluate the inhibition of metabolic bacterial activity, 20 μL of 2,3,5-triphenyl tetrazolium chloride (TTC) 0.125% (*w*/*v*) was added to all the wells, followed by 1 h of incubation. Tetrazolium salt is frequently employed in MIC determinations—when it is dissolved in water, it is colorless, but turns red when metabolically active bacteria are present. The red color is directly correlated with the number of living cells. The MBC was determined by seeding 20 μL from all clear MIC wells onto Mueller–Hinton Agar (MHA, Millipore Sigma) plates and was defined as the lowest concentration of EOs that killed 99.9% of the inoculum. Data from at least three replicates were evaluated and modal results were calculated.

#### 4.3.3. Checkerboard Assay

The checkerboard assay was used to determine potential synergistic, additive, or even antagonistic effects of combinations of EOs (*O. vulgare* L./*C. ladaniferus* L., *O. vulgare* L./*C. auratium* L. var. *amara* and *O. vulgare* L./*J. communis* L.). Dilutions of two EOs in combination, from 2 x MIC to the serial dilution below, were inoculated in microtiter plates and incubated as described above [59]. The checkerboard test was used to calculate the fractional inhibitory concentration (FIC) according to the following formulas: FICA = MICA + B/MICA; FICB = MICB + A/MICB; and FIC Index = FICA + FICB (where MICA + B is the MIC of compound A in the presence of compound B, and MICB + A is the opposite).

FIC Index (FICI) values were interpreted as follows: synergistic effect for FICI ≤ 0.5; additive effect for FICI > 0.5–1; indifference for FICI > 1 to <2; and antagonism for FICI ≥ 2 [19]. All experiments were performed in triplicate. The results were also reported as isobolograms, constructed by plotting synergistic concentrations [60].

#### 4.3.4. Effect on Biofilm Formation

The effect of EO combinations on the biofilm-forming ability of *S. aureus* ATCC 6538 was tested on polystyrene, flat-bottomed microtiter plates, as previously described [61]. The overnight culture in TSB (Millipore Sigma) + 1% glucose (TSBG) of *S. aureus* was adjusted in the TSBG to 1 × 10^6^ CFU/mL and was dispensed into each well of 96-well polystyrene flat-bottomed microtiter plates containing twofold dilutions of the EO combinations from the 1/4/ + 1/4 combination. The final concentration of bacteria was equal to 5 × 10^5^ CFU/mL and 5 × 10^4^ CFU/well, approximately. After incubation at 37 °C for 24 h, the planktonic phase was removed and each well was washed twice with sterile PBS (pH 7.4), dried, stained for 1 min with 0.1% safranin and washed with water. The stained biofilm biomass was re-suspended in 30% (*v*/*v*) acetic acid and OD_492_ was measured using a spectrophotometer EIA reader. A biofilm control consisting of a TSBG medium was included. The reduction percentage of the biofilm was calculated using the following equation:100 − (mean OD_492_ of EO association/mean OD_492_ of control well) × 100(1)

### 4.4. Antioxidant Activity

Individual EOs were screened for the antioxidant activity of the tested EOs using the following assays: 2,2-diphenyl-1-picrylhydrazyl radical (DPPH); stable 2,2′-azino-bis(3-ethylbenzthiazoline-6-sulfonic acid (ABTS); and ferric reducing antioxidant power (FRAP). Measurements were obtained in triplicate for each sample in each assay. The EC_50_ values were calculated for the control and samples, representing the antioxidant capacity in the sample necessary for 50% of the maximal antioxidant effect.

#### 4.4.1. 2,2-Diphenyl-1-picrylhydrazyl (DPPH) Test

The free radical-scavenging capacity of EOs was determined by the 2,2-diphenyl-1-picrylhydrazyl (DPPH) assay [62], a method based on a reduction of the stable DPPH radical. The reagent mixture consisted of 1.5 mL of 100 mM DPPH in methanol, to which 37.5 µL of solutions containing various concentrations (100–1000 mg/mL) of the EOs to be tested, or of the vehicle alone (DMSO), were added; an equal volume of the solvent employed to dissolve the extracts was added to control tubes. After 20 min of incubation at room temperature, the absorbance was recorded at 517 nm using a UV–Vis spectrophotometer. Trolox reagent was used as blank. Each determination was carried out in triplicate.

#### 4.4.2. 2,2′-Azino-bis-(3-ethylbenzothiazoline-6-sulfonic Acid) (ABTS) Assay

This method is used to determine the capacity of the EOs to quench the azino-bis-(3-ethylbenzothiazoline-6-sulfonic acid) radical (ABTS·^+^). In our experiments, the ABTS·^+^ radical cation was produced by the oxidation of 1.7 mM ABTS with potassium persulfate (4.3 mM final concentration) in water, according to Chelly et al. [63]. The mixture was allowed to stand in the dark at room temperature for 12–16 h before use. Then, the ABTS·^+^ solution was diluted with phosphate-buffered saline (PBS) at a pH of 7.4 to obtain an absorbance of 0.7 ± 0.02 at 734 nm. One hundred microliters of a solution containing different concentrations (1000–100 mg/mL) of the EO samples to be tested or of the vehicle alone (DMSO) were added to 2 mL of the ABTS·^+^ solution; the absorbance was recorded at 734 nm in a UV–Vis spectrophotometer after allowing the reaction to stand for 6 min in the dark at room temperature. Trolox reagent was used as blank. Each determination was carried out in triplicate.

#### 4.4.3. Ferric Reducing/Antioxidant Power (FRAP) Assay

The ferric reducing ability of the EOs under study was evaluated according to the method described by Chelly et al. [64], with minor modifications. The FRAP reagent contained 10 mM of TPTZ solution in 40 mM HCl, 20 mM FeCl_3_·6H_2_O, and acetate buffer (300 mM, pH 3.6) (1:1:10, *v*/*v*/*v*). A total of 50 μL of a methanolic solution containing different concentrations (100–1000 mg/mL) of the samples tested or of the vehicle (methanol) alone were added to 3 mL of the FRAP reagent; the absorbance was measured at 593 nm after incubation at 20 °C for 4 min using the FRAP reagent as a blank.

### 4.5. Anti-Inflammatory Activity

The in vitro anti-inflammatory activity of the EOs was evaluated according to the method of Belkhodja et al. [65] by monitoring the inhibition of protein denaturation. The method consisted of preparing 0.5 mL of a reaction mixture consisting of 0.45 mL BSA (5% aqueous solution) and 0.05 mL of EOs (250 μg/mL). A standard mixture of diclofenac (0.5 mL) was prepared in the same conditions (0.45 mL BSA 5% and 0.05 mL of the standard solution of diclofenac with a concentration of 10–100 μg/mL). The pH was calibrated at 6.3 using 1N HCl. After preparation, mixtures were incubated at 37 °C for 20 min and subsequently heated at 57 °C for 30 min. After cooling the samples, 2.5 mL of phosphate-buffered saline (pH 6.3) was added to each test tube. Moreover, 0.05 mL distilled water was used in place of essential oil in the control test tube while the product control did not contain bovine serum albumin. The absorbance was measured by a UV–Vis spectrophotometer (Shimadzu UV-1280, Shimadzu, Milan, Italy) at 416 nm and the inhibition percentage of the protein denaturation was calculated.

### 4.6. Antioxidant and Anti-Inflammatory Activities of EOs Combinations

The antioxidant and anti-inflammatory activities of the the following EO binary combinations were also evaluated: O. *vulgare* L./*C. ladaniferus* L.; O. *vulgare* L./*J. communis* L.; and O. *vulgare* L./*C.auratium* L. var. *amara*. For determining the types of interactions within the EOs, different doses of EO were combined in a ratio of 1:1. Evaluations of the different types of interactions (synergism, antagonism, or additive effect) between the EOs in binary combinations were carried out by transforming the experimental data studies to fractional inhibitory concentration (FIC) values. The fractional inhibitory concentration fifty percent indexes (FIC_50_I) were determined for each EOS combination according to Sharma et al. [39].

#### Fractional Inhibitory Concentration (FIC)

The sum of the fractional inhibitory concentration index (ΣFIC) was used to measure the interactions of different combinations of EO in a ratio of 1:1 when tested using the DPPH, FRAP, and ABTS assays. The same calculation was used to evaluate the anti-inflammatory capacity of the combined EOs through the inhibition of albumin denaturation.

The ΣFICs for each of the combinations were calculated using the following Equations:FIC(I) = EC_50_ (*a*) in combination with (*b*)/EC_50_ (*a*) independently(2)FIC(II) = EC_50_ (*b*) in combination with (*a*)/EC_50_ (*b*) independently(3)
where (*a*) is the EC_50_ of one EO in the combination and (*b*) is the EC_50_ of the other EO.

The ∑FICs for each combination were interpreted as having synergy when the ∑FICs were less than or equal to 0.5, as having an additive effect when the ∑FICs were greater than 0.5 but less than or equal to1.0, as having an indifferent effect when the ∑FICs were greater than 1.0 but less than or equal to 4.0, and as having an antagonistic effect when the ∑FICs were greater than 4.0.

### 4.7. Statistical Analysis

The results were statistically analyzed by one-way or two-way analysis of variance (ANOVA) tests, followed by Tukey’s honestly significant difference test, using the statistical software ezANOVA (https://people.cas.sc.edu/rorden/ezanova/index.html accessed on 1 September 2024).

## 5. Conclusions

This study highlighted the antibacterial, antioxidant and anti-inflammatory activities of the EOs of *C. ladaniferus* L., *C. aurantium* L. var. *amara, J. communis* L. and *O. vulgare* L., alone and in binary combinations. Their possible use in the treatment of animal infectious diseases caused by *S. aureus*, including drug-resistant and biofilm-forming strains, was investigated. Among all the EOs, *O. vulgare* L. was shown to be the most effective in enhancing the antibacterial, anti-biofilm, antioxidant, and anti-inflammatory activities of the other EOs when used in combination. Specifically, synergistic and additive effects were observed for *O. vulgare* L./*C. ladaniferus* L. and *O. vulgare* L./*J. communis* L. against *S. aureus* and MRSA, respectively. The additive effects observed for antioxidant and anti-inflammatory activities are also very important in mitigating infectious diseases associated with oxidative stress and inflammation, such as mastitis and mammary pustular dermatitis, which compromise animal health and production. In addition, the synergistic effects of these EO combinations could be an important tool in the food industry for ensuring the safety and preservation of foods of animal origin.

## Figures and Tables

**Figure 1 molecules-30-00438-f001:**
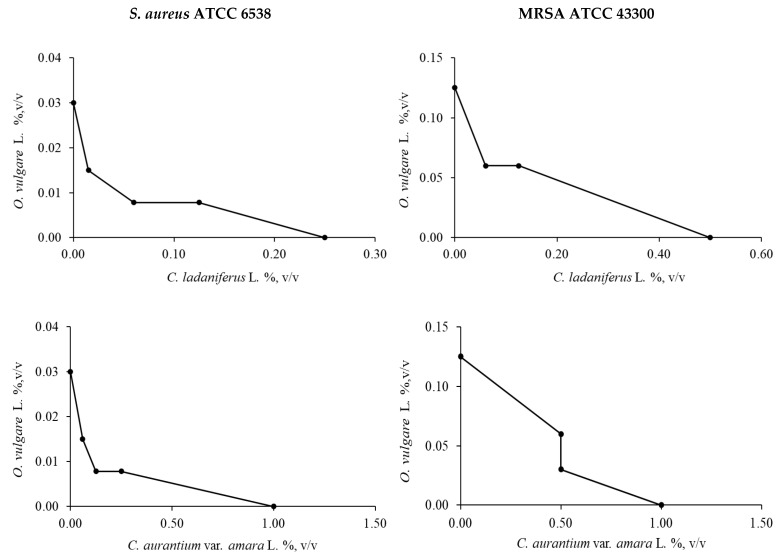
Isobolograms depicting the effect of *O. vulgare* L. in combination with *C. ladaniferus* L., *C. aurantium* var. *amara* L. and *J. communis* L. against *S. aureus* ATCC 6538 and MRSA ATCC 43300, determined by the checkerboard test and calculation of the fractional inhibitory concentration (FIC).

**Figure 2 molecules-30-00438-f002:**
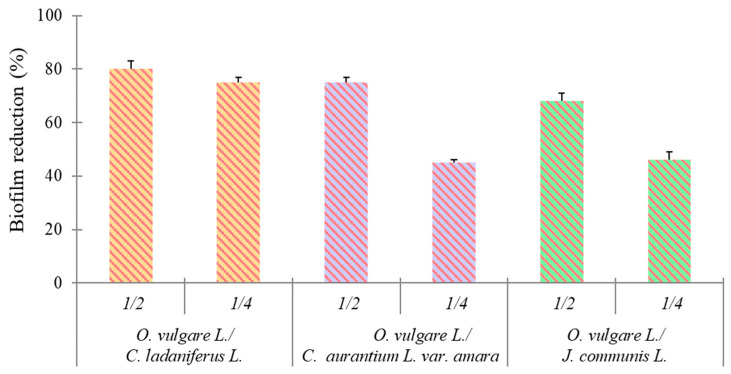
Reduction in the biofilm of *S. aureus* ATCC 6538 in the presence of sub-synergistic concentrations of 1/4 + 1/4 EOs combination.

**Figure 3 molecules-30-00438-f003:**
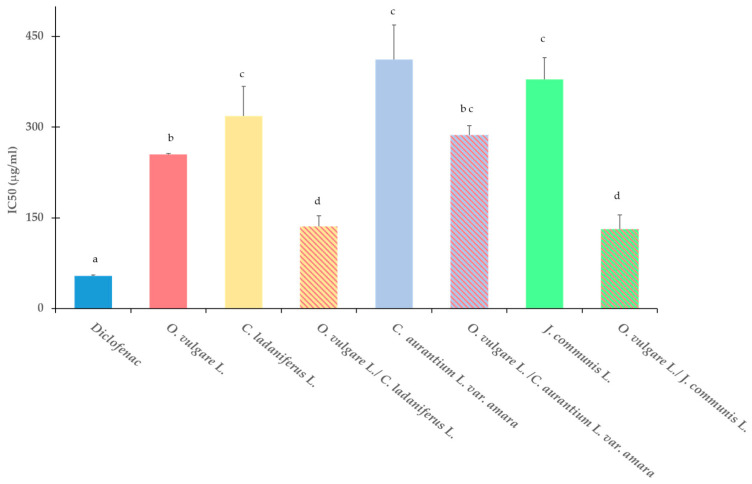
Effect on heat-induced protein denaturation of EOs alone and in combination (1:1), expressed as IC_50_ (inhibitory concentration 50%). Standard anti-inflammatory drug: diclofenac (sodium salt). Results are expressed as means ± SD of three different experiments. Means with the same letter are not significantly different from each other (*p* > 0.05).

**Table 1 molecules-30-00438-t001:** Antibacterial activity of the tested EOs.

Strains	*Cistus* *ladaniferus* L.		*Citrus auratium* L. var. *amara*		*Juniperus communis* L.		*Origanum* *vulgare* L.	
	MIC	MBC	MIC	MBC	MIC	MBC	MIC	MBC
	(%, *v*/*v*)							
*S. aureus* ATCC 6538	0.25	0.5	1	>1	1	>1	0.03	0.06
*S. aureus* (MRSA) ATCC 43300	0.5	1	1	>1	1	>1	0.12	0.25
*E. coli* ATCC 10536	0.5	1	1	>1	1	>1	0.06	0.12

**Table 2 molecules-30-00438-t002:** Antibacterial activities of *O. vulgare* L. in combination with the other EOs, determined by the checkerboard test and a calculation of the fractional inhibitory concentration (FIC) and fractional inhibitory concentration index (FICI).

Strains	Checkerboard	Best Combination ^a^		
		FIC	FICI	Effect
*S. aureus* ATCC 6538	*O. vulgare* L./*C. ladaniferus* L.	0.250/0.250	0.5	Synergy
	*O. vulgare* L./*C. aurantium* L. var. *amara*	0.250/0.125	0.375	Synergy
	*O. vulgare* L./*J. communis* L.	0.250/0.062	0.312	Synergy
*S. aureus* (MRSA) ATCC 43300	*O. vulgare* L./*C. ladaniferus* L.	0.5/0.125	0.625	Additive
	*O. vulgare* L./*C. aurantium* L. var. *amara*	0.250/0.5	0.75	Additive
	*O. vulgare* L./*J. communis* L.	0.5/0.5	1	Additive
*E. coli* ATCC 10536	*O. vulgare* L./*C. ladaniferus* L.	1/0.5	1.5	Indifference
	*O. vulgare* L./*C. aurantium* L.var. *amara*	1/0.5	1.5	Indifference
	*O. vulgare* L./*J. communis* L.	1/0.25	1.25	Indifference

^a^ Best combination of sub-MICs of EOs yielding the lowest FICI.

**Table 3 molecules-30-00438-t003:** Antioxidant activity of the tested EOs in three different in vitro redox-based assays.

Species	EC_50_ (μg/mL)		
	DPPH	ABTS	FRAP
*C. ladaniferus* L.	^a^ 804 ± 49	^a^ 647 ± 86	^a^ 954 ± 51
*C. aurantium* L. var. *amara*	^b^ 924 ± 26	^b^ 1077 ± 65	^b^ 1062 ± 54
*J. communis* L.	^c^ 720 ± 90	^c^ 786 ± 19	^c^ 729 ± 51
*O. vulgare* L.	^d^ 188 ± 38	^d^ 407 ± 72	^d^ 556 ± 64
Trolox	150 ± 36	61 ± 8	
FeSO_4_			67 ± 10

For each assay, means with the same letter are not significantly different from each other (*p* > 0.05).

**Table 4 molecules-30-00438-t004:** Antioxidant activities of the combinations (1:1 *v*/*v*) of EOs in the DPPH, ABTS, and FRAP assays.

Combinations	EC_50_ (μg/mL)		
	DPPH	ABTS	FRAP
*O. vulgare* L./*C. ladaniferus* L.	^a^ 144 ± 33	^a b^ 291 ± 69	^a^ 339 ± 27
*O. vulgare* L./*C. aurantium* L. var. *amara*	^b^ 275 ± 25	^b^ 366 ± 50	^b^ 421 ± 31
*O. vulgare* L./*J. communis* L.	^c^ 208 ± 22	^a^ 265 ± 31	^a^ 299 ± 22

For each assay, means with the same letter are not significantly different from each other (*p* > 0.05).

**Table 5 molecules-30-00438-t005:** Effect of EOs in combination in DPPH, ABTS, FRAP assays, expressed as the sum of the fractional inhibitory concentration index (ΣFIC).

Combination	DPPH		ABTS		FRAP	
	ΣFIC	Effect	ΣFIC	Effect	ΣFIC	Effect
*O. vulgare* L./*C. ladaniferus* L.	0.945	Additive	1.165	Indifference	0.964	Additive
*O. vulgare* L./*C. aurantium* L. var. *amara*	1.758	Indifference	1.240	Indifference	1.154	Indifference
*O. vulgare* L./*J. communis* L.	1.390	Indifference	0.989	Additive	0.947	Additive

**Table 6 molecules-30-00438-t006:** Effects of EOs associations in BSA denaturation assay, expressed as the sum of the fractional inhibitory concentration index (ΣFIC).

Combination	BSA	
	ΣFIC	Interaction
*O. vulgare* L./*C. ladaniferus* L.	0.964	Additive
*O. vulgare* L./*C. aurantium* L. var. *amara*	1.774	Indifference
*O. vulgare* L./*J. communis* L.	0.862	Additive

## Data Availability

All data and results related to this study are included in the article.

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
