# Peer review of "Binary Combinations of Essential Oils: Antibacterial Activity Against Staphylococcus aureus, and Antioxidant and Anti-Inflammatory Properties"

_molecules, 2025, doi:10.3390/molecules30030438_

Round 1
Reviewer 1 Report
Comments and Suggestions for Authors
The manuscript, entitled "Binary Combinations of Essential Oils: Antibacterial Activity Against Staphylococcus aureus, and Antioxidant and Anti-inflammatory Properties", publishes research on the antibacterial properties of essential oils. The article is written in clear language, the narrative is consistent, the conclusions are confirmed by the results. The scientific work was done competently, the literary sources were well used in discussing the results, and their list is sufficient. Essential oils have been a well-known remedy for a long time, but the composition of these oils differs in different types of plants. This article is one of many works devoted to the study of essential oils of plants. The topic of the article has no scientific novelty, but the work is of practical interest, since antibiotic resistance is a problem and we need to find alternative ways to combat bacterial infections. The authors were able to achieve success in the fight against staphylococcal and intestinal infections, including biofilms, and prepared practical recommendations. The article is complete, carries valuable information and is recommended for adoption after minor changes. 1) Pages 3 and 15. There is too much free space at the end of the page that is not occupied by the text. 2) On the tables and figures, the text is written in different fonts that differ from the font of the text of the main article. It seems that the article uses three or four different fonts. It is advisable to use one font. 3) Table 1 is really big. First, it can be divided into 4 tables, by plant species. Secondly, it can be removed in additional materials or in appendices to facilitate the perception of the article. They do not carry key information. 4) Figure 4. The captions are too small at the bottom, it is advisable to enlarge the font. 5) Tables 2, 3, 4, 5, 6 and 7. It would be better to align the first column with the names of the species, on the left edge. 6) Chapters 4.4.1 and 4.4.2. In the name of chemicals, a dot is placed after "2,2-". Perhaps this point is superfluous. Check if the spelling is correct.
Author Response
Reviewer #1
Comments and Suggestions for Authors
The manuscript, entitled "Binary Combinations of Essential Oils: Antibacterial Activity Against Staphylococcus aureus, and Antioxidant and Anti-inflammatory Properties", publishes research on the antibacterial properties of essential oils. The article is written in clear language, the narrative is consistent, the conclusions are confirmed by the results. The scientific work was done competently, the literary sources were well used in discussing the results, and their list is sufficient. Essential oils have been a well-known remedy for a long time, but the composition of these oils differs in different types of plants. This article is one of many works devoted to the study of essential oils of plants.
The topic of the article has no scientific novelty, but the work is of practical interest, since antibiotic resistance is a problem and we need to find alternative ways to combat bacterial infections. The authors were able to achieve success in the fight against staphylococcal and intestinal infections, including biofilms, and prepared practical recommendations. The article is complete, carries valuable information and is recommended for adoption after minor changes.
RE: We thank the Reviewer for the very positive feedback on our work and for his/her effort in reviewing the manuscript.
1) Pages 3 and 15. There is too much free space at the end of the page that is not occupied by the text.
- The manuscript has been formatted and the spaces on page 3 and 15 have been removed, as indicated.
2) On the tables and figures, the text is written in different fonts that differ from the font of the text of the main article. It seems that the article uses three or four different fonts. It is advisable to use one font.
- The manuscript was revisited accordingly, and the same fonts were used throughout the text.
3) Table 1 is really big. First, it can be divided into 4 tables, by plant species. Secondly, it can be removed in additional materials or in appendices to facilitate the perception of the article. They do not carry key information.
- Table 1 was divided into four different sections based on plant species, moved to the Supplementary Material file and renamed Table S1.
4) Figure 4. The captions are too small at the bottom, it is advisable to enlarge the font.
- Done.
5) Tables 2, 3, 4, 5, 6 and 7. It would be better to align the first column with the names of the species, on the left edge.
- Done.
6) Chapters 4.4.1 and 4.4.2. In the name of chemicals, a dot is placed after "2,2-". Perhaps this point is superfluous. Check if the spelling is correct.
- Done.
Reviewer 2 Report
Comments and Suggestions for Authors
Abstract: What is new? Highlight this!
References: Far too many references, several seems to be of minor significance. Less than half the number is required. Use primary references as much as possible. Secondary refr are normally used in the discussion section as they normally support and add to primary findings. Check editional issues with refr 20. Refr: Is this a book? Editors, page(s)?
Introduction: Very plain and general text. Be more concise: A brief background followed by a description of the scientific status: What is known in this field, which gaps in knowledge exist, and finally describe how this contribution plans to fill these gaps.
Results: What is new in section 2.1? If the section doesn't make any scientific contribution then leave it out. Section 2.2 seems to be more relevant. Print error in line 141 (section 2.1.1)? Check line 166.
Fig 1 and Tab 1 should be published as supplemental material as there is no new scientific contribution here. Quantitative data would be more interesting in this context.
Fig 2: Check scaling of the axes! Zoom in!
Fig 4: Difficult to read. Enhance font size?
Discussion: line 234-243 This is not a discussion of results, rather introduction material. Remove.
Author Response
Reviewer #2
Comments and Suggestions for Authors
Abstract: What is new? Highlight this!
- The topic of this manuscript is certainly not new, but the work may be of practical interest, considering that antibiotic resistance is a worldwide problem that needs to be addressed according to a One-Health Moreover, the use of EOs as therapeutic adjuvants could be a viable, simple, and inexpensive strategy to combat bacterial infections. The novelty of this study was to test novel binary associations between known and uncommon EOs, and the results demonstrated their efficacy for potential treatment of staphylococcal infections, including those biofilm-associated. As suggested, the abstract was revised and the novelty of the study was highlighted.
References: Far too many references, several seems to be of minor significance. Less than half the number is required. Use primary references as much as possible. Secondary refr are normally used in the discussion section as they normally support and add to primary findings.
- The references were revisited throughout the text; those of minor significance were removed as suggested, but others were added in response to other reviewers’ requests.
Check editional issues with refr 20. Refr: Is this a book? Editors, page(s)?
- Corrected accordingly.
Introduction: Very plain and general text. Be more concise: A brief background followed by a description of the scientific status: What is known in this field, which gaps in knowledge exist, and finally describe how this contribution plans to fill these gaps.
- The introductory section was modified and some sentences were revised based on the comments and suggestions of all four reviewers.
Results: What is new in section 2.1? If the section doesn't make any scientific contribution then leave it out. Section 2.2 seems to be more relevant.
- Section 2.1 has been significantly shortened and the quantitative results have been removed as they are detailed in the Supplementary Material file.
Print error in line 141 (section 2.1.1)?
- Sorry for the mistake. It’s section 2.2.1
Check line 166.
- Done.
Fig 1 and Tab 1 should be published as supplemental material as there is no new scientific contribution here. Quantitative data would be more interesting in this context.
- The modification was made accordingly.
Fig 2: Check scaling of the axes! Zoom in!
- Done.
Fig 4: Difficult to read. Enhance font size?
- Done.
Discussion: line 234-243 This is not a discussion of results, rather introduction material. Remove.
- The sentence is a summary of results obtained in this study. However, it has been modified based on comments and suggestions from all reviewers.
Reviewer 3 Report
Comments and Suggestions for Authors
Minor comments:
1. Line 107: it should be 1.84% rather than 1,84%
2. Line 128-129L: Correct the spelling and punctuation in the sentences (Linear retention). Also, these could be added as a footnote for the table.
3. Line 130, 131, 132, and 133: Use Italic style, or be consistent in styling your binomial names.
4. Line 149: Describe briefly the classification limits for your FIC/FICI, before presenting the results.
5. Line 194: Correct for/from
6. Discuss in more details why you have selected these specific combination percentages for synergy in antioxidant and anti-inflammatory effects.
7. The author mentioned many times that lowering the dose can reduce side effects. To strengthen your points and enhance the significance of the results I would recommend adding a paragraph that give more specific details about this: Add in the discussion what type of side effects are expected or known for these essential oils/or their components upon application (internally or externally). Specify the dose that can cause these side effects. And then, compare with new doses that you have recommended for synergy.
8. Correct the figure number Line 366
Author Response
Reviewer #3
Comments and Suggestions for Authors
Minor comments:
- Line 107: it should be 1.84% rather than 1,84%.
- Revised accordingly. However, it has been removed from the main text as the detailed composition of EOs are now reported in a Supplementary Material file based of other reviewers’ requests.
- Line 128-129L: Correct the spelling and punctuation in the sentences (Linear retention). Also, these could be added as a footnote for the table.
- The sentences have been corrected accordingly and moved to the Supplementary Material file.
- Line 130, 131, 132, and 133: Use Italic style, or be consistent in styling your binomial names.
- Revised accordingly and moved to the Supplementary Material file.
- Line 149: Describe briefly the classification limits for your FIC/FICI, before presenting the results.
- As suggested, classification limits for FIC/FICI have been included before presenting the results as follows: The Fractional Inhibitory Concentration (FIC) was calculated according to the formulas: FICA = MICA + B/MICA, FICB = MICB + A/MICB, and FIC Index = FICA + FICB, where MICA + B is the MIC of compound A in presence of compound B, and MICB + A is the opposite. FIC Index (FICI) values were interpreted as follows: synergistic effect FICI ≤ 0.5; additive effect FICI >0.5 - ≤ 1; indifference FICI >1 - < 2; antagonism FICI ≥ 2.
- Line 194: Correct for/from .
- Corrected accordingly.
- Discuss in more details why you have selected these specific combination percentages for synergy in antioxidant and anti-inflammatory effects.
- The combinations were chosen, both taking into consideration other articles in literature, i.e. DOI:10.1371/journal.pone.0131321, and the effective concentrations obtained by testing the EOs individually.
- The author mentioned many times that lowering the dose can reduce side effects. To strengthen your points and enhance the significance of the results I would recommend adding a paragraph that give more specific details about this: Add in the discussion what type of side effects are expected or known for these essential oils/or their components upon application (internally or externally). Specify the dose that can cause these side effects. And then, compare with new doses that you have recommended for synergy.
- The main purpose of this study was to find effective binary associations of EOs to reduce the doses to be used. Indeed, although EOs are natural products, it is known that they can be responsible for several toxic side-effects (Wojtunik-Kulesza, K.A. Toxicity of Selected Monoterpenes and Essential Oils Rich in These Compounds. Molecules 2022, 27, 1716, doi:10.3390/molecules27051716) mainly because of their chemical complexity. To date, it is difficult to define a real safety profile for EOs, hence, the possibility of combining them in binary associations allows for a reduction in the dose required for activity and thus the risk of potential side effects. As suggested this point has been added in the discussion.
- Correct the figure number Line 366.
- Sorry for the mistake. The figure number has been corrected as Figure 5.
Reviewer 4 Report
Comments and Suggestions for Authors
Dear authors,
I have made some corrections and included questions in the text. In addition to that, I have the following suggestions:
Combine Table 1 into a single table for all oils, but only include the most dominant components. Large and detailed tables like this can be moved to the supplementary materials.
Figure 3. add control.
Why did not you analyze the effect of individual oils on biofilm formation?
Arrange the references as per guidelines, and pay attention to the formatting of Latin names.

Author Response
Reviewer #4
Comments and Suggestions for Authors
Dear authors,
I have made some corrections and included questions in the text:
Line 140 and 170:
- Sorry for the mistake. Cistus Ladaniferus has been corrected as Cistus ladaniferus L.
Line 201:
- “The ability to inhibit” has been corrected as “The ability in inhibit”.
Line 236:
- The sentence has been corrected as suggested.
Line 305:
- Sorry for the mistake. Cistus Ladaniferus has been corrected as Cistus ladaniferus L.
Line 314:
- Positive control (ciprofloxacin) was included.
Line399:
- The bacterial concentration in the well was 5 × 104 CFU/well, approximatively.
Line 404:
- It was measured the absorbance at 492 nm using a microplate reader.
Line 427:
- The final concentration of bacteria was equal to 5 × 105 CFU/mL and 5 × 104 CFU/well, approximatively.
Line 478:
- The comma has been added.
Line 538-791:
- The italics has been corrected for the botanical names. Sorry for the mistake. All the references have been reconciled.
In addition to that, I have the following suggestions:
Combine Table 1 into a single table for all oils, but only include the most dominant components. Large and detailed tables like this can be moved to the supplementary materials.
- The table in question has been reorganized and moved to the Supplementary Material file as per other reviewer’s suggestion.
Figure 3. add control.
- ‘ The OD492 of aureus ATCC 6538 biofilm control stained with safranin was 1.50 ± 0.15.’ This sentence has been added in the text.
Why did not you analyze the effect of individual oils on biofilm formation?
- Individual EOs were not analyzed because the aim of this study was to evaluate the biofilm inhibition of the binary association of oregano oil, known for its antibiofilm activity (Nostro, A. et al. Effects of Oregano, Carvacrol and Thymol on Staphylococcus aureus and Staphylococcus epidermidis J. Med. Microbiol. 2007, 56, 519–523, doi:10.1099/jmm.0.46804-0.), with the other EOs.
Arrange the references as per guidelines and pay attention to the formatting of Latin names.
- Done.

Round 2
Reviewer 2 Report
Comments and Suggestions for Authors
Comments:
Tab 1: Significant numbers in last two columns? I recommend to include until hundredths.
Fig 1: Shorten the x- and y-axes (zoom in).
Tab 3 and 4: Significant numbers? Remove decimals!
Author Response
Reviewer#2
Comments and Suggestions for Authors
Comments:
Tab 1: Significant numbers in last two columns? I recommend to include until hundredths.
- Corrected as requested.
Fig 1: Shorten the x- and y-axes (zoom in).
- Figure 1 has been modified and zoomed, as requested.
Tab 3 and 4: Significant numbers? Remove decimals!
- The numbers in Table 3 and 4 have been corrected as requested.
Reviewer 4 Report
Comments and Suggestions for Authors
Reference no.6 change Aureus to aureus.
Author Response
Reviewer#4
Comments and Suggestions for Authors
Reference no.6 change Aureus to aureus.
- Sorry for the mistake. The reference n.6 has been corrected.